# User Preference-aware Fake News Detection

Yingtong Dou[1], Kai Shu[2], Congying Xia[1], Philip S. Yu[1], Lichao Sun[3]
[1]Department of Computer Science, University of Illinois at Chicago, Chicago, IL, USA
[2]Department of Computer Science, Illinois Institute of Technology, Chicago, IL, USA
[3]Department of Computer Science and Engineering, Lehigh University, Bethlehem, PA, USA
{ydou5,cxia8,psyu}@uic.edu,kshu@iit.edu,james.lichao.sun@gmail.com

## ABSTRACT

Disinformation and fake news have posed detrimental effects on individuals and society in recent years, attracting broad attention to fake news detection. The majority of existing fake news detection algorithms focus on mining news content and/or the surrounding exogenous context for discovering deceptive signals; while the endogenous preference of a user when he/she decides to spread a piece of fake news or not is ignored. The *confirmation bias* theory has indicated that a user is more likely to spread a piece of fake news when it confirms his/her existing beliefs/preferences. Users' historical, social engagements such as posts provide rich information about users' preferences toward news and have great potentials to advance fake news detection. However, the work on exploring user preference for fake news detection is somewhat limited. Therefore, in this paper, we study the novel problem of exploiting user preference for fake news detection. We propose a new framework, UPFD, which simultaneously captures various signals from user preferences by joint content and graph modeling. Experimental results on real-world datasets demonstrate the effectiveness of the proposed framework. We release our code and data as a benchmark for GNN-based fake news detection: https://github.com/safe-graph/GNN-FakeNews.

## CCS CONCEPTS

• **Computing methodologies** → *Machine learning*; • **Information systems** → **Social networks**.

## KEYWORDS

Data Mining; Social Media Analysis; Fake News Detection

**ACM Reference Format:**
Yingtong Dou[1], Kai Shu[2], Congying Xia[1], Philip S. Yu[1], Lichao Sun[3]. 2021. User Preference-aware Fake News Detection. In *Proceedings of the 44th International ACM SIGIR Conference on Research and Development in Information Retrieval (SIGIR '21), July 11–15, 2021, Virtual Event, Canada.* ACM, New York, NY, USA, 5 pages. https://doi.org/10.1145/XXXXXX.XXXXXX

## 1 INTRODUCTION

In recent years, social media has enabled the wide dissemination of disinformation and fake news– false or misleading information disguised in news articles to mislead consumers [27, 37]. Disinformation has resulted in deleterious effects and raised serious concerns, demanding novel approaches for fake news detection.

Among various fake news detection techniques, fact-checking is the most straightforward approach; however, it is usually labor-intensive to acquire evidence from domain experts [9]. In addition, computational approaches using feature engineering or deep learning have shown many promising results [3, 12, 23, 31]. For example, SAFE [36] and FakeBERT [11] used the TextCNN [35] and BERT [5, 28, 29] to encode the news textual information, respectively; GCNFN [17] and GNN-CL [8] leveraged the GCN [14] to encode the news propagation patterns on social media (e.g., news sharing cascading among social media accounts). However, these methods focus on modeling news content and its user exogenous context and ignore the user endogenous preferences.

Sociological and psychological studies on journalism have theorized the correlation between user preferences and their online news consumption behaviors [24]. For example, *Naïve Realism* [22] indicates that consumers tend to believe that their perceptions of reality are the only accurate views, while others who disagree are regarded as uninformed, irrational, or biased; and *Confirmation Bias* theory [18] reveals that consumers prefer to receive information that confirms their existing views. For instance, a user believes the election fraud would probably share similar news with a supportive stance, and the news asserting election is stolen would attract users with similar beliefs [1]. To model user endogenous preferences, existing works have attempted to utilize historical posts as a proxy and have shown promising performance to detect sarcasm [13], hate speech [20], and fake news spreaders [21] on social media.

In this paper, we consider the historical posts of social media users as their endogenous preference in news consumption. We propose an end-to-end fake news detection framework named User Preference-aware Fake Detection (UPFD) to model endogenous preference and exogenous context jointly (as shown in Figure 1). Specifically, UPFD consists of the following major components: (1) To model the user endogenous preference, we encode news content and user historical posts using various text representation learning approaches. (2) To obtain the user exogenous context, we build a tree-structured propagation graph for each news based on its sharing cascading on social media. The news post is regarded as the root node, and other nodes represent the users who shared the same news posts. (3) To integrate the endogenous and exogenous information, we take the vector representations of news and users as their node features and employ Graph Neural Networks (GNNs) [7, 17] to learn a joint user engagement embedding. The user engagement embedding and news textual embedding are used to train a neural classifier to detect fake news. Our major contributions can be summarized as follows:

*SIGIR '21, July 11–15, 2021, Virtual Event, Canada.*
© 2021 Association for Computing Machinery.
ACM ISBN 978-1-4503-8037-9/21/07...$15.00
https://doi.org/10.1145/XXXXXX.XXXXXX

Yingtong Dou[1], Kai Shu[2], Congying Xia[1], Philip S. Yu[1], Lichao Sun[3]

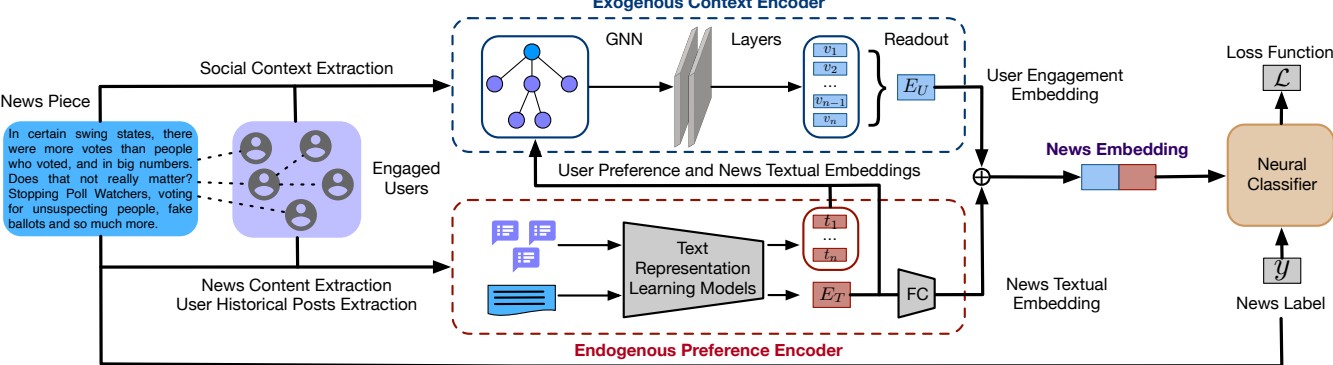

**Figure 1: The proposed** UPFD **framework for user preference-aware fake news detection. Given the news piece and its engaged users on social media, we extract the exogenous context as a news propagation graph and encode the endogenous information based on user historical posts and news texts. The endogenous and exogenous information are fused using a GNN encoder. The final news embedding, composed of user engagement embedding and news textual embedding, is fed into the neural classifier to predict the news' credibility.**

- We study a novel problem of user preference-aware fake news detection on social media;
- We propose a principled way to exploit both endogenous preference and exogenous context jointly to detect fake news; and
- We conduct extensive experiments on real-world datasets to demonstrate the effectiveness of UPFD for detecting fake news.

## 2 OUR APPROACH

In this section, we present the details of the proposed framework for fake news detection named UPFD (User Preference-aware Fake News Detection). As shown in Figure 1, our framework has three major components. First, given a news piece, we crawl the historical posts of the users engaged in the news to learn user endogenous preference. We implicitly extract the preferences of engaged users by encoding historical posts using text representation learning techniques (e.g., word2vec [16], BERT [5]). The news textual data is encoded using the same approach. Second, to leverage user exogenous context, we build the news propagation graph according to its engagement information on social media platforms (e.g., retweets on Twitter). Third, we devise a hierarchical information fusion process to fuse the user endogenous preference and exogenous context. Specifically, we obtain the user engagement embedding using GNN as the graph encoder, where the news and user embeddings encoded by the text encoder are used as their corresponding node features in the news propagation graph. The final news embeddings are composed by the concatenation of user engagement embedding and news textual embedding.

Next, we will introduce how we encode endogenous preference, extract the exogenous context, and fuse both information.

### 2.1 Endogenous Preference Encoding

It is non-trivial to explicitly model the endogenous preference of a user only using his/her social network information. Similar to [2, 13, 20] which model the users' personality, sentiment and stance using their historical posts, we leverage the historical posts of a user to encode his/her preference implicitly. However, none of the previous fake news datasets contain such information. In this paper, we select

the FakeNewsNet dataset [25] which contains news content and its social engagement information on Twitter. Then we use the Twitter Developer API [4] to crawl historical tweets of all accounts that retweeted the news in FakeNewsNet.

To obtain rich historical information for user preference modeling, we crawl the recent two hundred tweets for each account, so as to near 20 million tweets being crawled in total. For inaccessible users whose accounts are suspended or deleted, we use randomly sampled tweets from accessible users engaging the same news as its corresponding historical posts. Because deleting the inaccessible users will break the intact news propagation cascading and results in a less effective exogenous context encoder. We also remove the special characters, e.g., "@" characters and URLs, before applying text representation learning methods.

To encode the news textual information and user preferences, we employ two types of text representation learning approaches based on language pretraining. Instead of training on the local corpus, the word embeddings pretrained on a large corpus are supposed to encode more semantic similarities between different words and sentences. For pretrained word2vec vectors, we choose the 680k 300-dimensional vectors pretrained by spaCy [10]. We also employ pretrained BERT embeddings to encode the historical tweets and news content as a sequence [5] using bert-as-a-service [32].

Next, we elaborate the details of applying the above text representation learning models. spaCy includes pretrained vectors for 680k words, and we average the vectors of existing words in combined recent 200 tweets to get user preference representation. The news textual embedding is obtained similarly. For the BERT model, we use the cased BERT-Large model to encode the news and user information. The news content is encoded using BERT with maximum input sequence length (i.e., 512 tokens). Due to BERT's input sequence length limitation, we could not use BERT to encode 200 tweets as one sequence, so we resort to encode each tweet separately and average them afterward to obtain a user's preference representation. Generally, the tweet text is way shorter than the news text, we empirically set the max input sequence length of BERT as 16 tokens to accelerate the tweets encoding time.

## 2.2 Exogenous Context Extraction

Given a news piece on social media, the user exogenous context is composed of all users that engaged with the news. We utilize the retweet information of news pieces to build a news propagation graph. As the toy example of the news propagation graph shown in Figure 1, it is a tree-structured graph where the root node represents the news piece, and other nodes represent users who share the root news. In this paper, we investigate the fake news propagation on Twitter as a proof-of-concept use case. To build propagation networks in Twitter, we follow a similar strategy used in [8, 17, 26]. Specifically, we define a new piece as $v_1$, and $\{v_2, \ldots, v_n\}$ as a list of users that retweeted $v_1$ ordered by time. We define two following rules to determine the news propagation path:

- For any account $v_i$, if $v_i$ retweets the same news later than at least one following accounts in $\{v_1, \ldots, v_n\}$, we estimate the news spreads from the account with the latest timestamp to account $v_i$. Since the latest tweets are first presented in the timeline of the Twitter app, and thus have higher probabilities to be retweeted.
- If account $v_i$ does not follow any accounts in the retweet sequences including the source account, we conservatively estimate the news spreads from the accounts with the most number of followers. Because tweets from accounts with more followers have a higher chance to be viewed/retweeted by other users according to the Twitter content distributing rules.

Based on the above rules, we can build the news propagation graphs on Twitter. Note that this approach can be applied to other social media platforms like Facebook as well.

## 2.3 Information Fusion

Previous works [8, 15, 17] have demonstrated that fusing the user features with a news propagation graph could boost the fake news detection performance. Since the GNN can encode both node feature and graph structure in an end-to-end manner, it is a good fit for our task. Specifically, we propose a hierarchical information fusion approach. We first fuse the endogenous and exogenous information using the GNN. With a GNN, the news textual embedding and user preference embedding can be taken as node features. Given the news propagation graph, most GNNs aggregate the features of its adjacent nodes to learn the embedding of a node. Like previous GNN-based graph classification models [33, 34], we apply a readout function over all node embeddings to obtain the embedding of a news propagation graph. The readout function makes the mean pooling operation over all node embeddings to get the graph embedding (i.e., user engagement embedding). Second, since the news content usually contains more explicit signals regarding the news' credibility [3]. We fuse the news textual embedding and user engagement embedding by concatenation as the ultimate news embedding to enrich the news embedding information.

The fused news embedding is finally fed into a two-layer Multi-layer Perceptron (MLP) with two output neurons representing the predicted probabilities for fake and real news. The model is trained using a binary cross-entropy loss function and is updated with SGD.

## 3 EXPERIMENTS

In the experiments, we want to address two following questions: **RQ1:** How are the performances of the proposed UPFD framework

**Table 1: Dataset and graph statistics.**

| Dataset | #Graphs (#Fake) | #Total Nodes | #Total Edges | #Avg. Nodes per Graph |
|---|---|---|---|---|
| Politifact (POL) | 314 (157) | 41,054 | 40,740 | 131 |
| Gossipcop (GOS) | 5464 (2732) | 314,262 | 308,798 | 58 |

compared to previous works? **RQ2:** What are the contributions of endogenous/exogenous information and other variants of the proposed framework?

### 3.1 Experimental Setup

*3.1.1 Dataset.* Previous works have proposed a couple of fake news datasets with news pieces from different websites and their fact-checking information [19, 30]. To investigate both the user preference and propagation pattern of fake news, we choose the Fake-NewsNet dataset [25]. It contains fake and real news information from two fact-checking websites and the related social engagement from Twitter. The dataset statistics are shown in Table 1.

*3.1.2 Baselines.* We compare the UPFD with fake news detection models that utilize different information. Many baseline methods leverage extra information like image information which is not included in the FakeNewsNet [25]. To ensure a fair comparison, we implement the baselines only with the parts for encoding the news content, user comments, and news propagation graph. CSI [23] employs an LSTM to encode the news content information to detect fake news. SAFE [36] uses the TextCNN [35] to encode the news textual information. GCNFN [17] is the first fake news detection framework to encode the news propagation graph using GCN [14]. It takes the profile information and comment textual embeddings as the user feature. GNN-CL [8] encodes the news propagation graph using DiffPool [34], a GNN designed for graph classification. The node features are extracted from user profile attributes on Twitter. The list of ten profile feature names can be found in [8, 15] We also add two baselines that apply MLP directly on news textual embeddings encoded by word2vec and BERT.

*3.1.3 Experimental Settings.* We implement all models using Py-Torch, and all GNN models are implemented with PyTorch-Geometric package [6]. We use unified graph embedding size (128), batch size (128), optimizer (Adam), and L2 regularization weight (0.001), train-val-test split (20%-10%-70%) for all models. The experimental results are averaged over five different runnings. Other hyper-parameters for each model are reported with the code.

### 3.2 RQ1: Performance Evaluation

Table 2 shows the fake news detection performance of UPFD and six baselines. First, we can observe that UPFD has the best performance comparing to all baselines. UPFD outperforms the best baseline GCNFN around 1% on both datasets with statistical significance. The experimental results of UPFD and GCNFN demonstrate that the user comments (used by GCNFN) are also beneficial to fake news detection; and the user endogenous preference could impose additional information when user comment information is limited.

Yingtong Dou[1], Kai Shu[2], Congying Xia[1], Philip S. Yu[1], Lichao Sun[3]

**Table 2: The fake news detection performance of baselines and our model. Stars denote statistically significant under the t-test ($* \ p \leq 0.05, ** \ p \leq 0.01, *** \ p \leq 0.001$).**

| | Model | POL | | GOS | |
|---|---|---|---|---|---|
| | | ACC | F1 | ACC | F1 |
| News Only | SAFE [36] | 73.30 | 72.87 | 77.37 | 77.19 |
| | CSI [23] | 76.02 | 75.99 | 75.20 | 75.01 |
| | BERT+MLP | 71.04 | 71.03 | 85.76 | 85.75 |
| | word2vec+MLP | 76.47 | 76.36 | 84.61 | 84.59 |
| News + User | GNN-CL [8] | 62.90 | 62.25 | 95.11 | 95.09 |
| | GCNFN [17] | 83.16 | 83.56 | 96.38 | 96.36 |
| | UPFD (ours) | 84.62* | 84.65* | 97.23** | 97.22*** |

Second, since all baselines either encode the news content or user comments without considering the historical posts, we can tell that leveraging the historical posts as user endogenous preferences could improve the fake news detection performance. Note that the UPFD with the best performance on the both datasets uses BERT as the text encoder and GraphSAGE as the graph encoder.

**Table 3: Fake news detection performance on two datasets with different node feature types and models. The bold (underlined) text indicates the best (second best) performances on each dataset.**

| Feature | POL | | | | GOS | | | |
|---|---|---|---|---|---|---|---|---|
| | GraphSAGE | | GCNFN | | GraphSAGE | | GCNFN | |
| | ACC | F1 | ACC | F1 | ACC | F1 | ACC | F1 |
| Profile | 77.38 | 77.12 | 76.94 | 76.72 | 92.19 | 92.16 | 89.00 | 88.96 |
| word2vec | 80.54 | 80.41 | 80.54 | 80.41 | 96.81 | 96.80 | 94.97 | 94.95 |
| BERT | **84.62** | **84.53** | 83.26 | 83.14 | **97.23** | **97.22** | 96.18 | 96.17 |

### 3.3 RQ2: Ablation Study

*3.3.1 Encoder Variants.* As we mentioned in Section 2.3, we employ different text encoders and GNNs to encode the endogenous and exogenous information. In Table 3, we show the fake news detection performance of two GNN variants using three different node features. Note that "word2vec" and "BERT" represent features encoding the user endogenous preferences while the "Profile" feature is regarded as a baseline. GraphSAGE [7] is a GNN to learn node embeddings via aggregating neighbor nodes information and GCNFN [17] is a GNN-based fake news detection model which leverages two GCN layers to encode the news propagation graph.

Table 3 shows that the endogenous features (word2vec and BERT) are consistently better than the profile feature, which only encodes the user profile information. We also observe that GraphSAGE and BERT have the average best performance among other model and feature variants. It suggests that BERT is better than word2vec for encoding textual features which has been verified on other NLP tasks [5]. Note that the BERT performance could be further improved via fine-tuning, and we leave it as future work.

*3.3.2 Framework Variants.* To verify the effectiveness of endogenous preference and exogenous context, we fix the text and graph encoder and design three UPFD variants that remove the endogenous information, exogenous information or both of them.

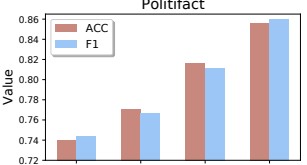 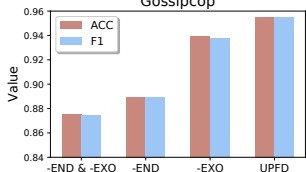

**Figure 2: The fake news detection performance of different variants of UPFD framework. -END/-EXO represents the UPFD variant without endogenous/exogenous information.**

Specifically, we employ the GCNFN (word2vec) as the graph (text) encoder for both datasets, and remove news concatenation to ensure a fair comparison. The UPFD variant without exogenous information (-EXO) is implemented by removing all edges in the news propagation graph. Thus, -EXO encodes the news embedding solely based on node features without exchanging information between nodes. The UPFD variant without endogenous information (-END) takes the user profile as node features and does not contain user endogenous preference information. The UPFD variant without both endogenous and exogenous information (-END & -EXO) replaces the node features of the -EXO with user profile features.

Figure 2 shows the fake news detection performance for different UPFD variants on two datasets. We can find that removing either component from the UPFD will reduce its performance. Moreover, jointly encoding the endogenous and exogenous information attains the best performance. The accuracy of UPFD/-EXO is 85.61%/81.63%, and the F1 score of UPFD/-EXO is 85.97%/81.15% on Politifact. The accuracy of UPFD/-EXO is 95.47%/93.92%, and the F1 score of UPFD/-EXO is 95.46%/93.81% on Politifact. All the experimental results are statistically significant under the t-test ($p \leq 0.01$). This indicates that exogenous information (i.e., news propagation graph) is more informative on Politifact since removing it results in a larger performance drop. It is obvious that endogenous information contributes more to performance gain than exogenous information. This observation further verifies the necessity of modeling user endogenous preferences.

### 4 CONCLUSION

In this paper, we argues that user endogenous news consumption preference plays a vital role in the fake news detection problem. To verify this argument, we collect the user historical posts to implicitly model the user endogenous preference and leverage the news propagation graph on social media as the exogenous social context of users. An end-to-end fake news detection framework named UPFD is proposed to fuse the endogenous and exogenous information and predict the news' credibility on social media. Experimental results demonstrate the advantage of modeling the user endogenous preference.

### ACKNOWLEDGMENTS

This work is supported in part by NSF under grants III-1763325, III-1909323, and SaTC-1930941. Kai Shu is supported by the John S. and James L. Knight Foundation through a grant to the Institute for Data, Democracy & Politics at The George Washington University.

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
