# OpenReview forum: "User Preference-aware Fake News Detection"
_ACM.org/SIGIR/Badging_

### Official Review · ~Andrew_Trotman1 · 2021-06-01
**Not yet ready for badging**

**Comment:**

We have examined the artefacts associated with this paper and we are not ready to badge the paper. Although the artefacts are online, there are, in the opinion of the reviewers, a number of issues that need to be addressed.

 1.   There is no documentation associated with the data. At a minimum (for Functional Artefacts badging) the authors should document what each file is. Where, in the pipeline, the file is from (is it the embeddings, the twitter IDs, or the graph). And what format the file is in.

 2.   We prefer (for Reusable and Available) documentation of the byte-level formats of the files. This might be a link to a page somewhere else that gives details of the file formats in sufficient detail that a programmer of a different language (such as C++) can extract the data from the files. It appears as though the files are in format used by a Python library. Consequently, programmers in other languages cannot be expected to be familiar with these file formats, and cannot even know if they are standard or author created.

 3.   It is not at all clear which git commit was used to generate the results in the paper. At a minimum (for Functional Artefacts) the authors should tag the GitHub repo at the commit used to obtain the results in the paper (or, if the same results are generated with the latest version, then tag that).

 4.   There is no documentation associated with the software. At a minimum (for Functional Artefacts) the authors should document the programs in the utils directory and in the gnn_model directory. What does each program do and how is it called (what are the options, etc.)? If usage can be obtained from a command line option then document that. But others should not need to read the source code to work out what each program does. Furthermore, installation instructions are very minimal and could be expanded to aid programmers in installing the software.

**Awarded Badges:**

["No Badges"]

---

### Official Review · Program_Chairs · 2021-07-09
**Not ready yet for badging**

**Comment:**

Dear Authors,

According to the review and also after discussion with the reviewers, your artifact is not ready yet for badging. The review below contains detailed motivations.

You may consider if you wish to re-submit, at a later stage, your artifact for badging. As explained in the SIGIR AEC guidelines (https://sigir.org/general-information/acm-sigir-artifact-badging/):

> Authors can apply for a given badge at most three times.
> Authors re-submitting an artifact are expected to provide a link to the
> previous submission within OpenReview and
>  the submission form contains a specific field for this purpose.

Should you consider re-submitting your artifact, it should be submitted as a new and separate submission, providing a link to this previous submission.

**Awarded Badges:**

["No Badges"]